# Analysis of Flammability and Smoke Emission of Plastic Materials Used in Construction and Transport

**DOI:** 10.3390/ma16062444

**Published:** 2023-03-18

**Authors:** Monika Borucka, Kamila Mizera, Jan Przybysz, Paweł Kozikowski, Agnieszka Gajek

**Affiliations:** Department of Chemical, Biological and Aerosol Hazards, Central Institute for Labour Protection-National Research Institute, Czerniakowska 16, 00-701 Warsaw, Poland

**Keywords:** polyurethane foams, thermal stability, fire effluents analysis, steady-state tube furnace, fire toxicity

## Abstract

This study provides valuable data on the specific toxic products that could be released from the commercially used, flexible polyurethane foams (FPUFs) during a fire. The steady-state tube furnace (Purser furnace) was used to generate combustion and thermal degradation products under different fire conditions. The concentrations of asphyxiates and irritant gases were determined using a Fourier transform infrared spectroscopy gas analyser. The volatile and semi-volatile organic compounds released in the fire effluents were collected using the solid-phase microextraction technique and identified by gas chromatography with a mass selective detector. In addition, the thermal stability of the FPUFs was evaluated by simultaneous thermal analysis. The cone calorimetry test was used to determine the flame retardancy of the selected materials. The obtained results show that the emission of carbon monoxide and hydrogen cyanide during the thermal degradation and combustion of the tested foams exceeded the permissible values and pose a serious threat to human life and health. Moreover, substituted benzenes, aldehydes, and polycyclic hydrocarbons were found in the released gases during all of the test conditions.

## 1. Introduction

Flexible polyurethane foams (FPUFs) have many advantages, such as low density, aging resistance, good elasticity, and easy moulding, and are widely used in many industries, including the construction, insulation, automotive, and furniture industries [1,2]. FPUFs are nitrogen-containing polymer materials synthesized from polyols, isocyanates, etc. [1,2,3].

According to fire statistics, the majority of fires occur in buildings. One of the reasons for the high number of casualties in building fires is that materials such as polyurethane foam generate many toxic products, which cause suffocation [4]. Research shows that about 35.4% of residential fire deaths are caused by the combustion of FPUFs [1,5]. The combustion of FPUFs is accompanied by pyrolysis to form a polyol liquid, which creates a pool fire and causes the fire to spread [6,7]. Moreover, when FPUF is ignited in the upper levels of a building, the dropping behaviour of the polyols produced by the combustion may ignite the fuel on the lower levels, leading to a secondary fire source [1].

A flame retardant is very often used to improve the fire-retardant properties of FPUFs. These compounds, in the event of a fire, can prevent diffusion and ensure an adequate evacuation time [8]. For the practical use of FPUFs, the flame-retardant properties must meet different requirements, depending on the national or international standards and their end applications. The fire resistance of polyurethane foam is achieved using reactive or additive flame retardants [9,10]. Unfortunately, despite the improvement in the fire resistance of FPUF materials as a result of fire resistance regulations, the fire effluent emitted during the thermal degradation and combustion of FPUFs remains a significant issue [4,11].

Recent years have seen an increase in research into new polyurethane with superior flame retardancy [4,10]. Liu et al. [12] designed thermoplastic polyurethane composites based on an aluminium hypophosphite-modified iron tailings system. The halogen-free phosphorous-sulphur ionic liquid was successfully added to the matrix of FPUF [13]. Oliwa et al. [14] investigated the effect of the type and amount of expandable graphite and blackcurrant pomace on the flammability and thermal stability of viscoelastic polyurethane foams. The flame-retardant and smoke-suppressant flexible polyurethane foams were designed and synthesized based on expandable graphite and novel liquid phosphorus-containing polyol [15,16]. Moreover, Yangui Chen et al. [2] developed the pyrolysis model, two ignition models, and the gas-phase combustion model of FPUF based on thermogravimetry and cone calorimeter experiments. However, the cited publications have predominantly evaluated the heat release rate and total heat release via cone calorimetry, and they have not analysed the toxic gases emitted during combustion. Therefore, analysis of the gases emitted during the combustion and thermal degradation of polyurethane materials should be carried out in order to understand the full complexity and actual risk of a fire.

The exposure and potential health effects of the smoke produced during the combustion of flame-retardant polymers are unknown, partially due to the lack of exposure data. Several investigations have examined the toxicity of the smoke produced during the combustion of flame-retardant polymers. W. Netkueakul et al. [17] investigated the effects of graphene nanoplatelet (GNP) in epoxy composite on the aerosol released from its combustion. Their results confirmed the potential health risks of the aerosol emissions from epoxy composites at their end-of-life via a combustion process and, at the same time, highlighted that the incorporation of GNP does not induce any novel or additive adverse effects on alveolar epithelial cells within 96 h of culture after exposure. However, D. Singh et al. [18], during an in vitro cellular toxicological evaluation of polyurethane thermoplastic enabled with carbon nanotubes (PU-CNT), confirmed that PU-CNT showed significantly higher cytotoxicity compared to PU, which could be attributed to its higher PAH concentration. These investigations are crucial and make the case that the presence of fire retardants in polymers can significantly affect the physicochemical and toxicological properties of the products released during thermal decomposition and combustion.

Fire effluent toxicity is highly dependent on both the fire scenario and the composition of the combustible material [19,20]. A fire is an uncontrolled, chaotic process in which conditions change rapidly. Depending on the fire’s conditions, many diverse chemical compounds are produced. Moreover, these products are released in varying amounts. The chemicals contained within fire effluents are divided into four groups: asphyxiates (carbon dioxide, hydrogen cyanide, carbon monoxide), irritants (ammonia, nitrogen oxides, hydrogen chloride, phenol, sulphur dioxide), allergens (isocyanates), and carcinogens (polycyclic aromatic hydrocarbons, dioxins, furans, certain heavy metals) [21]. In addition, the combustion of organic materials, particularly if it is incomplete, may also give rise to more complex chemicals in the smoke, which may include longer carbon chains and multiple-carbon rings [22,23].

The measures of fire safety and the assessment of the toxic effects of fires on humans are the key factors for assessing fire hazards. It is also important to assess the environmental impact of the toxic compounds in fire effluents. The interaction between ecology and the toxic effects of fire effluents is very important, complex, and may involve the study of food chains with several different trophic levels. This makes the tracing of toxicants to obtain reliable results a significant challenge. Moreover, the bench-scale test is a cheaper and less complex method than large-scale fire simulations [24].

The most commonly used ISO standard methods using bench-scale tests in the assessment of fire effluent toxicity [24,25,26,27] are the steady-state tube furnace (ISO 19700 [24]), smoke density chamber (ISO 5659 [25]), and cone calorimeter (ISO 5660 [26]) tests. The steady-state tube furnace was designed specifically for the assessment of smoke toxicity. This method has been accepted as a British standard (BS 7990:2003) and an IEC standard (IEC 60695-7-50) as a technique for the replication of real decomposition conditions and the analysis of toxic fire products, and it is currently in the process of gaining acceptance as an international standard [28]. The smoke density chamber was designed to assess the smoke generation and the cone calorimeter is a tool to investigate the flammability and burning behaviour of materials. However, the ISO 5659 [25] smoke chamber and the ISO 5660 [26] cone calorimeter tests have some limitations. In the smoke chamber, the combustion conditions vary during the course of the combustion as the resulting smoke gases fill the closed chamber, whereas the combustion conditions in a cone calorimeter test are always well ventilated using the normal test procedure [28].

Many studies have focused on the smoke toxicity of polyurethanes and their compositions. The tests have been performed based on ISO 19700 [24] using a steady-state tube furnace. Unfortunately, it is hard to find research that contains comprehensive information. In most studies, only inorganic gases have been determined [28,29,30,31,32,33]. These studies have also focused on products that are released during decomposition and combustion taking place under selected measurement conditions. For example, during pyrolyzed at 950 °C [29], combustion occurred at 825 [30,32] or 600 °C [33] in under-ventilated conditions.

In this study, the toxic products emitted in fire effluents during the thermal degradation and combustion of four commercially used, flexible polyurethane foams were determined. The steady-state tube furnace [24] was used to generate fire effluents from real fires under the following combustion scenarios: oxidative pyrolysis; well-ventilated flaming; small, vitiated fires; and post-flashover fires. The concentrations of asphyxiates and irritant gases, as well as light hydrocarbons, were determined using a gas analyser (Fourier transform infrared (FT-IR) spectroscopy). The volatile and semi-volatile organic compounds emitted in the fire effluents were collected and sampled using the solid-phase microextraction technique (SPME) and identified using gas chromatography with a mass selective detector (GC-MS). As each test run represented the burning behaviour of a particular fire stage, the results are more generally applicable to determining fire hazards than those of tests where only selected fire stages are analysed. Moreover, in this work, the thermal stability of the FPURs was evaluated by simultaneous thermal analysis (STA). In addition, the cone calorimetry test was used to investigate the material flammability and burning behaviour of the selected polyurethane materials.

## 2. Experimental Section

### 2.1. Materials

The commercially used, flexible polyurethane foams were purchased from Poland. FPUF_A (density 23 kg/m^3^) and FPUF_D (density 26 kg/m^3^) are type T flexible polyurethane foams, and FPUF_B (density 81 kg/m^3^) and FPUF_C (density 108 kg/m^3^) are rebound polyurethane foams (type R). The reaction to the fire class according to PN-EN 13501 (Polish standard for reaction to fire) classified the foams as class E, i.e., self-extinguishing and not spreading fire. In practice, this means that the selected polyurethane foam does not pose an additional hazard in the event of a fire.

### 2.2. Methods

The thermal stability of the materials was defined by simultaneous thermal analysis using 449F3 Jupiter from Netzsch, Selb, Germany. The tested 10 mg samples were heated from room temperature to 800 °C with a heating rate of 10 °C/min. The tests were carried out in an air atmosphere with a flow rate of 50 mL/min. The values presented in this article are the averages obtained for at least three samples from each flexible polyurethane foam.

The cone calorimeter (CC) (Fire Testing Technologies, East Grinstead, UK) tests were performed to investigate the burning behaviour of the polyurethane foams. The specimens (100 × 100 × 4 mm) were placed in an aluminium tray and irradiated horizontally at a heat flux of 35 kW/m^2^. Spark ignition was used to ignite the pyrolysis products. The test procedures were performed in accordance with ISO 5660-1 [26]. All samples were tested three times.

The main objective of this work was the determination of the asphyxiates (e.g., carbon monoxide, carbon dioxide, and hydrogen cyanide), irritants (e.g., ammonia, hydrogen chloride, and nitrogen oxides), light hydrocarbons, and other organic compounds in gases that can be evolved in the combustion and thermal degradation of the selected materials. The experiments were carried out in the steady-state tube furnace (Purser furnace, ISO 19700 [24]), which has been used specifically to generate products from real fires. This method was used to model a wide range of fire conditions by using different combinations of temperatures, non-flaming and flaming decomposition conditions, and different fuel/oxygen ratios in the tube furnace. These included the different types of fires, as detailed in ISO 19706 [27], Table 1.

A test run was only valid if the selected steady-state conditions were maintained for a period of at least 5 min during the test. If the ignition occurred during a non-flaming run, or failed to occur during a flaming run, then the furnace temperature was raised or lowered in 25 °C steps until the required behaviour was obtained. A new test run was then carried out with a fresh test specimen. For the flaming behaviour, it was also necessary to ensure that the primary air flow rates were correct, as specified in the ISO.

The concentrations of the selected released chemical compounds, i.e., carbon dioxide (CO_2_), carbon monoxide (CO), hydrogen cyanide (HCN), nitrogen dioxide (NO_2_), nitrogen oxide (NO), hydrogen chloride (HCl), ammonia (NH_3_), formaldehyde (HCHO), and light hydrocarbons, were determined using a gas analyser (Fourier transform infrared (FT-IR) spectroscopy) coupled with a computer system (Gasmet DX-4000 analyser).

In addition, the volatile and semi-volatile compounds released during the thermal degradation and combustion of the selected materials were analysed using a gas chromatograph (GC 7890 A firm Agilent Technologies, Hanover, USA) with a mass spectrometer (MSD 5975 firm Agilent Technologies, Hanover, USA). To achieve this goal, solid-phase microextraction (SPME) was used as a technique, which combines sampling and concentrating analyses, as well as introducing them to the chromatographic system [34,35].

During the steady-state period of the test run, the sample of effluent was taken from the mixing chamber by introducing the SPME device into the sampling ports with a fibre (Supelco, Bellefonte, USA). The carboxen/polydimethylsiloxane (CAR/PDMS) fibre coatings were used because they are recommended for the extraction of non-polar and polar analyses [36]. Before use, the fibres were conditioned in the injection port, according to the manufacturer’s instructions. After introducing the SPME syringe to the mixing chamber, the gaseous products of the thermal decomposition were sorbed on the SPME fibre. After collection (10 min), the SPME fibre was withdrawn from the chamber and desorbed immediately in the GC injector for analysis.

Chromatographic separation was achieved on an HP-5MS fused-silica capillary column (30 m × 0.25 mm × 0.25 μm film thickness) using helium as the carrier gas at 1 mL/min. The oven temperature was maintained at 40 °C for 10 min, increased by 5 °C/min to 240 °C, and held for 8 min. The GC injector port was 250 °C. The MSD was operated by electronic impact (70 eV) in scan mode (25–450 m/z).

Chromatographic peaks were identified through comparing the mass ions of each peak with the NIST MS Library. Based on the NIST library, only relationships with a probability higher than 85% agreement were considered. The chromatographic peak area of a specific compound is correlated linearly with its quantity; therefore, its concentration can be reflected by the peak area ratio. The summed identified peak areas were normalized to 100% and the relative abundance of a specific compound was reflected by its peak area ratio. The values presented in Table 5 are the averages obtained for at least three samples from each flexible polyurethane foam.

## 3. Results and Discussion

### 3.1. Thermal Properties

The thermal behaviour of the FPUFs was investigated by TGA and DTG, and the results are shown in Figure 1 and summarized in Table 2.

The thermal decomposition of polyurethanes depends on the number of urethane linkages and the content of aromatic moieties [37]. Depending on the tested foams, the occurrence of one or more stages of decomposition corresponded to the maximum rate of the degradation of hard segments and soft segments. 

One of the differences between the TGA profiles of the tested foams was the presence of weight loss at 165–230 °C for the FPUF_B and FPUF_D foams. The first step of the weight loss during the thermogravimetric analysis of the FPUF_D foam was probably related to the evaporation of the flame-retardant tris-β-chloropropyl phosphate (TCPP), whose flash point is at 218 °C and decomposition temperature is 244 °C [38]. The subsequent tests confirmed the presence of TCPP in that foam. The main stage of the degradation of FPUF_D occurred between 250 and 350 °C. The obtained results are consistent with those obtained by X. Liu et al. [39].

However, the first stage at low temperatures could be attributed to the moisture absorbed by the foam and the evaporation of a foaming agent and low molecular weight compounds, especially during the decomposition of FPUF_B.

The second main stage of the degradation of the tested materials at 277–284 °C could be attributed to thermolysis processes eliminating the bond types created by the reactions between the diisocyanate and polyol and releasing molecules from the original tolylene diisocyanate (TDI) and high-molecular-weight polyol. The molecules derived from TDI have relatively low molecular weights, and are thus gasified and escape from the remaining foam. The higher molecular weight material derived from the polyol was left behind.

During the third stage, occurring at 316–357 °C, the destruction of isocyanate and diphenylmethane took place [40,41].

The last steps of the decomposition of the tested FPUFs occurred at 533–545 and 678–729 °C and were related to the decomposition of the aromatic compounds. Moreover, these steps can be explained by the oxidative reaction of the double bonds in the chains of the used substrates in the tested foams [42].

### 3.2. Burning Behavior

Table 3 shows the results of the cone calorimeter tests, while Figure 2 shows the evolution of the heat release rate (HRR) with time for four different samples. The amount of energy released per unit surface area is often used to assess the risk of fire. HRR represents the maximum fire intensity and is employed as a proxy for the rate and extent of the fire propagation. Specifically, a high HRR is correlated with danger at the initial fire stage [43].

As can be seen in Figure 2, the HRR curves showed a different course. The initial peak in the HRR could be associated with the combustion of the molecules derived from TDI produced during the pyrolysis. The FPUFs decomposed from the top down; thus, the remaining unreacted foams served to insulate the collapsing foam from the bottom of the aluminium tray in contact with the substrate. As a result of that process, the initial HRR curves were very similar. After the foam fully collapsed, the very thin liquid layer of polyol-derived material was deposited on the bottom of the aluminium tray. This thin layer had a much higher thermal conductivity than the original foam. Moreover, the narrow depth ensured a nearly uniform temperature throughout the liquid. When the liquid was heated to a temperature sufficient to induce pyrolysis, the combustion gases supporting the second stage of burning were released. It is worth noting that the strong dependence on the sample substrate arose because the collapsed liquid was in intimate contact with the bottom of the highly thermally conductive aluminium tray, which was, in turn, in strong thermal contact with the underlying substrate [44]. In the cases of FPUF_B and FPUF_C, both rebound polyurethane foams (type R), the HRR curves showed a different course. These complex materials indicated a lower value of peak HRR, but they released heat for a long time. Therefore, the values for the total heat released were the same for both samples, and were seven times higher than the total heat released (THR) for FPUF_D. The reduction in the pHRR observed for the FPUF_B and FPUF_C foams was probably due to the reaction between the fire retardants and the TDI or related nitrogen rich volatiles. The reduction in the THR in the case of FPUF_D demonstrates the incomplete combustion of this material [45]. One of the reasons for the decrease may be the formation of char. MARHE is used as an index for the hazard of developing fires. The lower the value of MARHE, the better the fire performance [46]. The MARHE parameter has similar dependence to pHRR.

### 3.3. Determination of the Products Evolved during the Thermal Degradation and Combustion 

The test series was started under well-ventilated conditions. A number of preparatory tests were conducted, and it was found that steady flaming combustion was attained at the nominal 650 °C. It was also found that, during the tests on the FPUF_B, FPUF_C, and FPUF_D foams, when the primary air flow rate was 10 L/min, the oxygen depletion (D_O2_) calculated from the average percent oxygen concertation in the mixing and measurement chamber was >3.14 %. Therefore, the test was repeated with a primary air flow of 15 L/min. The primary flow rates for the under-ventilated tests were calculated from the oxygen consumption in the well-ventilated tests, as described in ISO 19700. The pyrolysis tests were carried out at 350 °C, which is the standard advance rate given in ISO 19700. In that test, flaming combustion did not occur.

The conducted tube furnace tests, the test conditions, and the analyses conducted in each test are given in Table 4 for the flexible polyurethane foams.

#### 3.3.1. Determination of the Asphyxiates and Irritants 

The concentrations of asphyxiates and irritant gases released during the thermal decomposition and combustion of the samples of flexible polyurethane foams were determined and illustrated as a function of time, as seen in Figure 3.

The concentration of chemical compounds released in the fire effluents depended on the conditions of the decomposition and the fire stages. Observing the course of the evolution of the fire effluents, it could also be seen that the manner of their emission depended on the type of material. 

During oxidative pyrolysis, all of the polyurethane foams began emitting carbon monoxide at about 250 s. The evolution of CO during the combustion of FPUF_B, FPUF_C, and FPUF_D was, however, more homogeneous, with a maximum seen at approx. 690 s. On the other hand, the evolution of CO during the decomposition of FPUF_A was more complex and heterogeneous. All of the tested foams released insignificant amounts of hydrogen cyanide in the test conditions.

During well-ventilated flaming, when the temperature of combustion was 650 °C, all of the tested polyurethane foams, with the exception of FPUF_B, showed similar behaviour. On the rejected curves, there were maximum emissions at about 100–200 s, and then the CO emission decreased. The FPUF_B foam showed different behaviour, with the CO emissions increasing successively during all of the tests. In turn, by analysing the course of HCN release, similarities could be seen in the case of FPUF_A-FPUF_C and FPUF_B-FPUF_D. 

The combustion that occurred during small, vitiated fires and post-flashover fires resulted in the release of significant amounts of CO and HCN. Moreover, the evolution of these gases was more complex and depended on the type of flexible polyurethane foam. 

In Figure 4, the total amounts of asphyxiates (carbon monoxide, hydrogen cyanide), and irritants (ammonia, hydrogen chloride, nitrogen oxides) emitted during the 5 min steady-state periods are summarized. 

During the oxidative pyrolysis of the tested foams, all of the samples released very low yields of CO, NOx, HCN, and HCl. The largest yields of CO were detected in the fire effluents formed when the decomposition of the tested samples took place during flame combustion with under-ventilated conditions (3a and 3b). Additionally, more CO was produced during the combustion of FPUF_A and FPUF_C than during the combustion of FPUF_B and FPUF_C. The yields of CO increased when burning the polymers, both with the use of flame retardants, especially acting in the gas phase [47], and with under-ventilated conditions [19].

A similar dependence was observed for hydrogen cyanide. Very low yields of HCN were detected during combustion flaming and increased with under-ventilated conditions for all of the tested materials. Under well-ventilated combustion conditions, almost all of the carbon in the polyurethane foams was oxidized to CO_2_, while most of the nitrogen from the FPUFs was released as N_2_. The small amounts of nitrogen from the FPUFs were oxidized to form NO_x_ (mostly NO and very little NO_2_). Under vitiated combustion conditions, the oxidation of the FPUFs became less efficient, so that a significant proportion of carbon fuel was released as CO and nitrogen fuel as HCN, NH_3_, and other nitriles [48,49].

The formation mechanism of NO_X_ from flexible polyurethane foams is very complex. Nitric oxide (NO) was detected in all of the tests. Moreover, the yield of NO was much higher when the combustion occurred at a higher temperature. Zevenhoven and Kilpinen [50] postulated that in an oxidative atmosphere, an increase in the temperature promotes the oxidation of HCN to NO. At the same time, the NO_x_ obtained reacts with the NCO radical to form N_2_O, which could be formed from the large amount of isocyanate groups present in the polyurethanes [51]. This mechanism can explain the fact that, during the combustion of the tested foams at 650 °C in oxidative conditions, significant amounts of N_2_O were formed.

Moreover, hydrogen chloride was released during the thermal degradation and combustion of the tested foams. The presence of chlorine was probably due to the addition of flame retardants to the tested materials and likely depended on the type of retardant used. However, the different foams emitted HCl differently. In the case of FPUF_A and FPUF_C, significant yields of HCl were obtained during pyrolysis and combustion with low ventilation. In turn, FPUF_B emitted the largest amount of HCl during combustion at 650 °C and good ventilation. The highest yields of HCl were detected during the combustion of FPUF_D. 

#### 3.3.2. Determination of Light Hydrocarbons

The main light hydrocarbons observed in the fire effluents obtained during the thermal degradation and combustion of the flexible polyurethane foams were methane, ethane, ethylene, propane, and hexane (Figure 5). Methane was found in the samples released from all of the tested flexible polyurethane foams. The highest yields of CH_4_ were observed when flaming combustion occurred at 650 °C and 825 °C in poorly ventilated conditions. Methane is most likely a product of the decomposition of oxygenated compounds, such as ketones and aldehydes [52]. Ethylene was not detected during pyrolysis. The yields of C_2_H_4_ increased with the temperature, from 650 to 825 °C, for all of the tested samples, with the exception of the FPUF_D. FPUF_D foam, which showed a different behaviour, emitting the least amounts of all hydrocarbons during combustion at 825 °C. This result was probably due to the composition of the FPUF_D foam. The retardants present in FPUF_D likely decreased the release of aromatic compounds, hydrocarbons, and HCN [51].

#### 3.3.3. Determination of Semi-Volatile Products in Gases Evolved during Combustion

More than 76 semi-volatile compounds were identified in the fire effluents released during the combustion and thermal degradation of the tested polyurethane foams. The highest number of products was detected during pyrolysis and flame combustion under poorly ventilation conditions for all of the foams. The main products identified in the fire effluents released during the ~5 min steady-state periods during different fire stages are summarized in Table 5.

During the pyrolysis of FPUF_A foam at 350 °C, the main identified compounds were acetic acid, benzene, phenylacetylene, naphthalene, and 2,4-diisocyanato-1-methylbenzene (TDI). These products were found in the gases released during the pyrolysis of all of the tested foams. Moreover, TDI was found in all test conditions. The yields of TDI decreased when the combustion occurred at higher temperatures. 

This phenomenon is confirmed by the existing literature [53], which indicates that, in temperatures around 300 °C, the breaking of the urethane bond (R-N-C-O=O-R) occurs, and the isocyanate (R-N=C=O) and polyol (HO-R) are released. Then, the isocyanate reacts through a rearrangement with itself to form various carbodiimide structures, and with the amine formed from the cleavage of the urethane bond to produce volatile polyureas. These released products are characterized as “yellow smoke”.

When the combustion of the tested foams occurred in higher temperatures, the fire effluents were found to contain benzene, benzonitrile, toluene, styrene, phenyl isocyanate 1-2-(1-methylethoxy)-propanol, and 1-propoxy-2-propanol. These products came from the degradation of TDI and urethane [51]. The maximum yields of these compounds were observed during combustion at 650 °C in small, poorly ventilated fires. When the temperature of the combustion was 825 °C, the concentration of these gases decreased. Oxygen-containing compounds, such as aldehydes and ketones, were also found in the fire effluents released during the combustion of the polyurethane foams, which came from the polyol and urethane.

All of the polyurethane foams, with the exception of FPUF_D, released similar products. In the case of FPUF_D, one of the main decomposition products was hydrogen chloride. It was the main product detected in the fire effluent during flaming combustion at 650 °C in well-ventilated conditions. Moreover, during the pyrolysis of FPUF_D, the flame-retardant tris-β-chloropropyl phosphate (TCPP) was found in the released gases. However, when the temperature of combustion increased, it was not detected. Therefore, it can be concluded that the hydrogen chloride came from the decomposition of TCPP.

Polycyclic aromatic hydrocarbons (PAHs) were also found in the fire effluents released during the thermal degradation and combustion of the polyurethane foams. Herrera et al. [54] proposed—as an explanation for the congener distribution—the theory that first a single aromatic ring (benzene) is formed, and then it begins growing to produce heavier PAHs. For this reason, the concentration of naphthalene was higher than acenaphthylene, whose concentration was higher than phenanthrene. The yields of PAHs increased when the temperature of combustion increased from 350 to 850 °C. Similar behaviour was observed when increasing the temperature during the thermal degradation of organic wastes [55]; the products, such as aliphatic hydrocarbons, evolved into light gases and aromatics, including polycyclic aromatic hydrocarbons [51].

## 4. Conclusions

Fire effluent typically comprises many compounds and particulates that are known to be harmful to people and the environment. In this study, a complex investigation was performed to determine the asphyxiates, irritants, light hydrocarbons, and volatile and semi-volatile organic compounds emitted in fire effluents during the thermal degradation and combustion of commercially used, flexible polyurethane foams. The proposed method was used to analyse the concentrations of the products of the thermal decomposition and combustion of FPUFs during different fire scenarios, including pyrolysis and well-ventilated and under-ventilated combustion.

The yields of most of the combustion products from the tested foams depended on the ventilation conditions. The obtained results showed that the CO, HCN, and HCl emissions during the combustion of the tested foam samples exceeded the permissible values and may pose a serious threat to human life and health during a fire. Large amounts of asphyxiating and irritating gases, as well as methane, ethane, and ethylene, were found in the fire effluents during combustion in under-ventilated conditions. The obtained results also confirmed that the addition of flame retardants containing chlorine atoms causes the emission of large amounts of hydrochloride, even during combustion in well-ventilated conditions.

Moreover, during all of the tested conditions, the fire effluents contained the very toxic toluene diisocyanate and other highly varied chemical compounds, such as volatile organic compounds and polycyclic hydrocarbons.

Knowledge of the products emitted, and their concentrations, during the combustion of commercially used, flexible polyurethane foams under different fire conditions is very important for estimating the fire safety of these materials. The results of this study could also be valuable for researchers developing new safety materials. 

## Figures and Tables

**Figure 1 materials-16-02444-f001:**
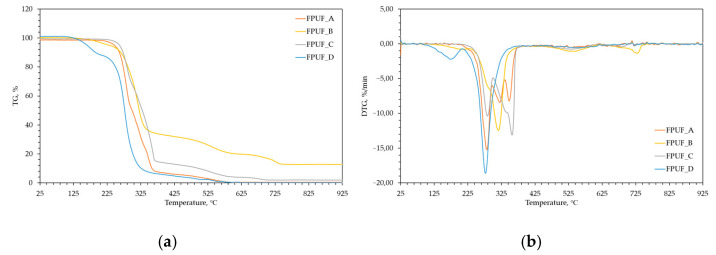
TGA (**a**) and DTG (**b**) curves of tested polyurethane foams.

**Figure 2 materials-16-02444-f002:**
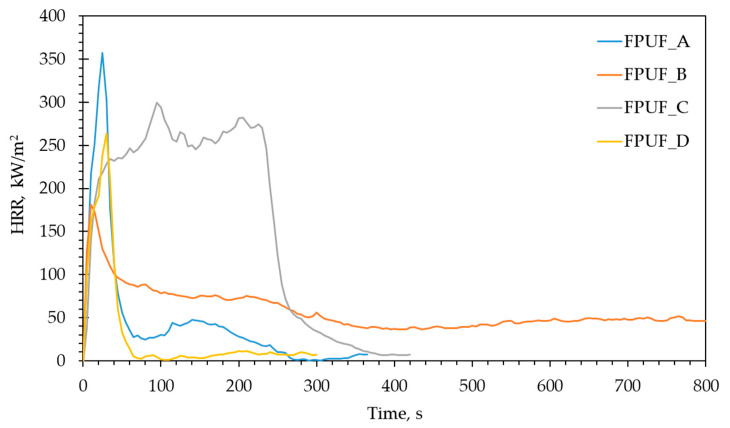
Representative heat release rate (HRR) curves of tested flexible polyurethane foams.

**Figure 3 materials-16-02444-f003:**
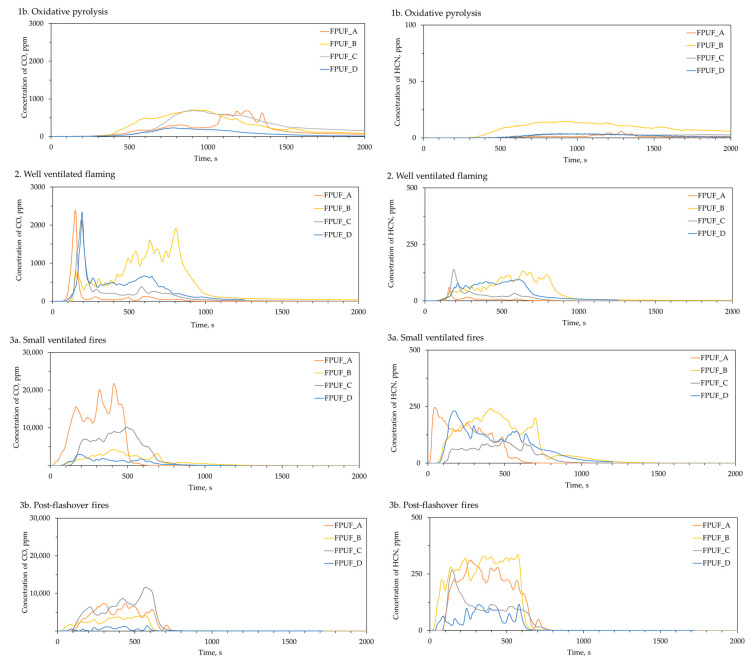
Yields of carbon monoxide (CO) and hydrogen cyanide (HCN) released during thermal degradation and combustion of flexible polyurethane foams in conditions presented in Table 4.

**Figure 4 materials-16-02444-f004:**
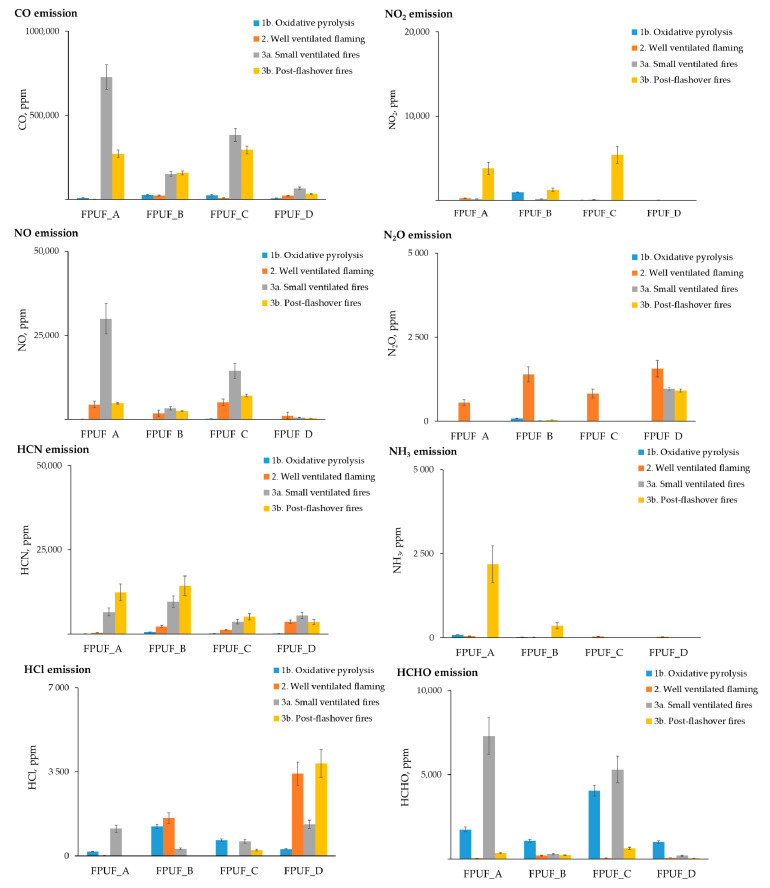
Total concentration of asphyxiates and irritants from steady-state tube furnace tests produced by flexible polyurethane foams during ~5 min steady-state periods.

**Figure 5 materials-16-02444-f005:**
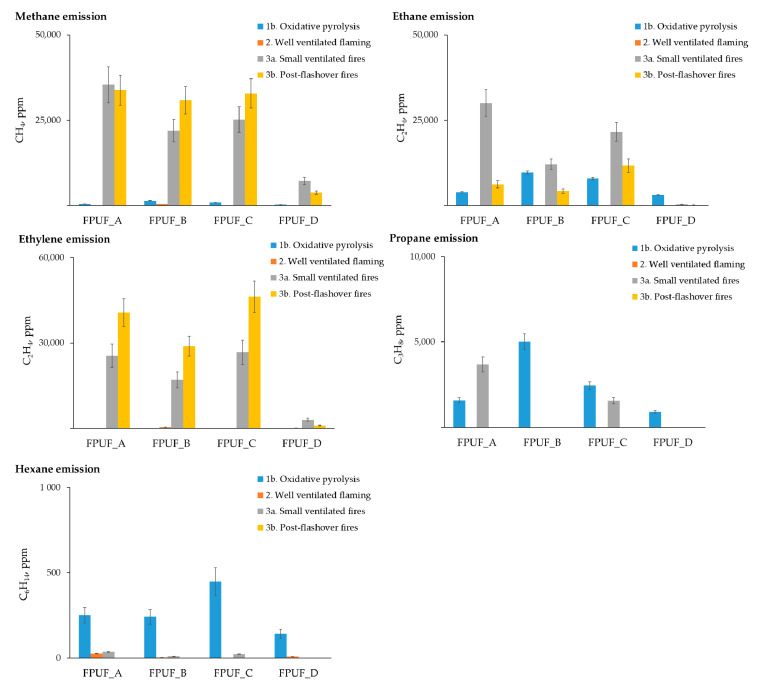
Total concentration of light hydrocarbons from steady-state tube furnace tests produced by flexible polyurethane foams during ~5 min steady-state periods.

**Table 1 materials-16-02444-t001:** ISO classification of fire stages, based on ISO 19706 [27].

Fire Stage		Max Temp., °C		Oxygen, %		Equivalence Ratio, ɸ	Combustion Efficiency, %
		Fuel	Smoke	In	Out		
Stage 1: Non-flaming	1b. Oxidative pyrolysis from externally applied radiation	300–600		20	20	−	50–90
Stage 2: Well-ventilated flaming (representing a flaming, developing fire)		350–650	50–500	~ 20	0–20	<0.75	>95
Stage 3: Less well-ventilated flaming	3a. Small, vitiated fires in closed or poorly ventilated compartments	300–600	50–500	15–20	5–10	2	70–80
	3b. Post-flashover fires in large or open compartments	350–650	600	<15	<5	2	70–90

**Table 2 materials-16-02444-t002:** Thermal properties of TGA and DTG analysis of tested flexible polyurethane foams.

Material	T_5%_, °C	T_1_, °C	V_1_, %/min	T_2_, °C	V_2_, %/min	T_3_, °C	V_3_, %/min	T_4_, °C	V_4_, %/min	T_5_, °C	V_5_, %/min	T_6_, °C	V_6_, %/min
FPUF_A	246 (1)			281 (1)	15.2 (0.26)	320 (0.5)	8.4 (0.25)	348 (3)	8.2 (0.24)				
FPUF_B	229 (1)	211 (0.5)	0.81 (0.05)			316 (1)	12.5 (0.28)			545 (5)	1.1 (0.14)	729 (2)	1.4 (0.26)
FPUF_C	263 (3)			284 (0.6)	10.4 (1.3)			357 (3)	13.1 (0.45)	533 (3)	0.78 (0.13)	698 (0)	0.68 (0.03)
FPUF_D	165 (1)	178 (0.6)	2.3 (0.12)	280 (2)	23.3 (0.15)								

**Table 3 materials-16-02444-t003:** Summary of cone calorimeter data.

Material	MP, g	MLR, g/s m^2^	TTI, s	pHRR, kW/m^2^	MAHRE kW/m^2^	THR, MJ/m^2^	TSR, m^2^/m^2^	CO, kg/kg	CO_2_, kg/kg
FPUF_A	6.06 (0.19)	3 (1)	3 (0.6)	358 (15)	233 (10)	17 (1)	204 (50)	0.025 (0.005)	1.21 (0.05)
FPUF_B	28.92 (0.5)	3 (0.6)	1 (0)	181 (10)	139 (13)	65 (5)	297 (28)	0.045 (0.01)	0.69 (0.04)
FPUF_C	26.05 (0.21)	10.8 (0.8)	4 (1)	299 (25)	246 (20)	65 (3)	1225 (52)	0.028 (0.025)	1.25 (0.1)
FPUF_D	7.67 (0.31)	2.51 (0.6)	2 (1)	264 (49)	177 (100)	9 (2)	411 (38)	0.072 (0.025)	0.52 (0.18)

**Table 4 materials-16-02444-t004:** Steady-state tube furnace test data for flexible polyurethane foams.

Sample	Test Conditions	Temp of Furnace, °C	Primary Air Flow Rate, L/min	Secondary Air Flow Rate, L/min	Total Mass Sample, g	Steady-State Period, s	Observations of Burning Behaviour
FPUF_A	1b. Oxidative pyrolysis	350	2	48	13.16	501–805	Non-flaming
	2. Well-ventilated flaming	650	10	40	14.81	237–541	Flaming D_O2_ = 2.47%
	3a. Small ventilated fires	650	3	47	16.15	171–475	Flaming
	3b. Post-flashover fires	825	3	47	17.34	303–607	Flaming
FPUF_B	1b. Oxidative pyrolysis	350	2	48	20.77	561–865	Non-flaming
	2. Well-ventilated flaming	650	15	35	20.49	125–429	Flaming D_O2_ = 3.14%
	3a. Small ventilated fires	650	3.8	46.2	20.57	251–555	Flaming
	3b. Post-flashover fires	825	3.8	46.2	20.95	251–555	Flaming
FPUF_C	1b. Oxidative pyrolysis	350	2	48	9.50	660–964	Non-flaming
	2. Well-ventilated flaming	650	15	35	10.88	290–594	Flaming D_O2_ = 3.14%
	3a. Small ventilated fires	650	3.8	46.2	9.18	251–555	Flaming
	3b. Post-flashover fires	825	3.8	46.2	9.64	185–488	Flaming
FPUF_D	1b. Oxidative pyrolysis	350	2	48	9.5	601–904	Non-flaming
	2. Well-ventilated flaming	650	15	35	8.03	270–540	Flaming D_O 2_ = 2.51%
	3a. Small, ventilated fires	650	3	47	8.57	303–607	Flaming
	3b. Post-flashover fires	825	3	47	9.64	303–607	Flaming

**Table 5 materials-16-02444-t005:** List of products identified in fire effluents released during ~5 min steady-state periods during different fire stages.

Detected Product	CAS	Amounts (%)															
		FPUF_A				FPUF_B				FPUF_C				FPUF_D			
		1B	2	3A	3B	1B	2	3A	3B	1B	2	3A	3B	1B	2	3A	3B
Oxalic acid	144-32-7									4.01 (1.6)							
Hydrogen chlorite	7647-01-0													5.84 (2.10)	72.76 (21.56)	14.91 (5.6)	19.84 (7.14)
Acrylonitrile	107-13-1	1.66 (0.82)		1.13 (0.52)	0.70 (0.10)	1.58 (0.63)	1.40 (0.34)	1.09 (0.25)	0.74 (0.17)			0.66 (0.23)	1.3 (0.5)	1.51 (0.78)		2.23 (1.14)	1.71 (0.87)
Acetic acid	64-19-7	14.82 (6.73)		1.42 (0.4)		2.02 (0.9)		1.12 (0.6)		2.79 (0.95)		1.61 (0.75)	1.09 (0.34)	2.87 (1.2)			
Crotonaldehyde	4170-30-3			0.33 (0.15)		0.20 (0.02)				0.18 (0.09)		1.61 (0.4)					
Hydroxyacetone	116-09-6									2.17 (1.1)				1.54 (0.7)			
Benzene	71-43-2	6.87 (2.1)	7.50 (3.2)	1.62 (0.62)	1.06 (0.42)	1.01 (0.45)		1.07 (0.5)	0.68 (0.43)			1.43 (0.54)	1.39 (0.6)			7.53 (3.5)	8.55 (4.3)
Butyronitrile	109-74-0	0.91 (0.43)		0.18 (0.05)													
Acetyl anhydride	108-24-7	1.12 (0.6)		0.40 (0.03)								0.39 (0.25)	0.12 (0.06)				
Hexan-2-ol	626-93-7	2.31 (1.3)		0.24 (0.1)		0.45 (0.03)				0.57 (0.25)				0.57 (0.2)			
Piridina	110-86-1			0.88 (0.3)	0.36 (0.13)				0.19 (0.1)			0.26 (0.1)	0.25 (0.05)				
Toluene	108-88-3		3.26 (1.21)	0.57 (0.05)	0.48 (0.12)			0.64 (0.32)	0.28 (0.08)			1 (0.4)	1.01 (0.54)				
2-Hydroxyethyl formate	628-35-3	1.59 (0.8)					0.36 (0.12)			0.53 (0.21)							
Diacetylamine	625-77-4	1.29 (0.65)											0.19 (0.05)				
Acetamide	60-35-5			0.71 (0.2)													
2,4-Pentadienenitrile	1615-70-9			0.79 (0.41)	0.50 (0.23)			0.40 (0.1)	0.27 (0.1)				0.12 (0.6)				
Propanoic acid, 2-oxo-, methyl ester	600-22-6	8.24 (2.6)				0.64 (0.34)				1.8 (0.8)				1.43 (0.7)			
1,2-Propanediol diformate	53818-14-7	2.92 (1.55)								0.46 (0.2)							
Pyridine, 2-methyl-	109-06-8			0.12 (0.03)													
Fumaronitrile	764-42-1			0.53 (0.14)													
1-(1-Methylethoxy)-2-propanone	42781-12-4	4.27 (2.1)		0.87 (0.38)		0.17 (0.05)				0.89 (0.5)		0.59 (0.3)		3.03 (1)			
3-Picoline	108-99-6			0.62 (0.4)		0.50 (0.1)											
Ethylbenzene	100-41-4											1.26 (0.5)					
Acetoxyacetone	592-20-1	1.45 (0.2)				0.74 (0.41)				2.43 (1.1)							
Phenylacetylene	536-74-3	12.36 (1.67)	2.36 (1.26)					0.34 (0.15)	0.24 (0.12)				0.21 (0.1)				
Styrene	100-42-5		6.01 (1.7)	0.75 (0.25)	1.37 (0.8)	3.53 (1.2)	1.38 (0.67)	3.93 (2)	1.03 (4.2)	3.24 (1.56)		9.49 (5.7)	4.15 (2.3)			1.57 (0.8)	1.99 (0.65)
Cumene	98-82-8											0.3 (0.1)					
Acetonylacetone	110-13-4	0.82 (0.25)		1.22 (0.64)		0.22 (0.1)				0.43 (0.2)		0.93 (0.4)					
Phenyl isocyanate	103-71-9			2.72 (1.2)	1.57 (0.9)	1.72 (0.56)	0.88 (0.5)	1.29 (0.4)	0.75 (0.35)								
Benzaldehyde	100-52-7		0.78 (0.25)	1.52 (0.7)		3.06 (1.24)	0.97 (0.43)	1.33 (0.3)		3.61 (1.7)		4.15 (2.10)	1.53 (0.76)			1.44 (0.62)	
2-Vinylpyridine	100-69-6												0.19 (0.1)				
3-Hexene-2,5-dione	4436-75-3													0.34 (0.13)			
N-Propylbenzene	103-65-1											0.2 (0.12)					
2-(2-Chloroethoxy)ethanol	628-89-7					0.29 (0.1)											
Aniline	62-53-3			0.59 (0.35	2.07 (1.1)				0.08 (0.05)				1.44 (0.6)				
2-Phenyl-1-propene	98-83-9					0.32 (0.15)		0.55 (0.1)									
Benzonitryle	100-47-0		4.98 (3.6)	27.24 (13.15)	9.17 (4.45)	0.32 (0.1)	13.29 (5.1)	10.79 (4.21)			3.44 (1.28)	8.53 (3.8)	5.3 (1.95)		11.69 (4.8)	12.72 (5.2)	7.80 (2.95)
Phenol	108-95-2		1.86 (0.88)			0.62 (0.42)	3.35 (1.15)		5.59 (2.6)	1.35 (0.75)		2.33 (1.3)	0.88 (0.3)			6.68 (2.85)	1.59 (0.8)
Acetophenone	98-86-2									2.29 (1.18)		1.44 (0.85)	0.32 (0.1)				
Benzophurane	271-89-6						0.96 (0.55)	0.44 (0.6)	0.17 (0.08)							2 (0.5)	
3-Cyanopyridine	100-54-9	1.46 (0.8)		1.87 (0.85)	0.81 (0.52)			0.45 (0.2)	0.55 (0.2)								
2-Methylphenyl isocyanate	614-68-6		0.88 (0.4)	7.21 (3.5)	1.11 (0.6)			2.03 (1.09)									
p-Tolylisocyanate	622-58-2		0.68 (0.23)	6.93 (4.2)	1.59 (0.25)		0.63 (0.2)	2.18 (1)	0.67 (0.14)								
1-methoxyindole	54698-11-2			0.75 (0.25)	0.26 (0.21)	0.29 (0.18)		0.51 (0.35)	0.32 (0.21)								
2-Pyridylacetonitrile	2739-97-1			0.62 (0.32)		0.32 (0.2)		0.18 (0.03)	0.07 (0.1)								
2-Cyanophenol	611-20-1					0.34 (0.1)	0.83 (0.5)	0.85 (0.32)	0.26 (0.13)								
Pyrrole-2-carbonitrile	4513-94-4	0.70 (0.2)	1.26 (1)	1.00 (0.4)	0.54 (0.3)											0.48 (0.21)	
3-[3-(1-Methylethoxy)propoxy]-1-propanol	54518-03-5									1.98 (0.98)				2.37 (1.18)			
Naphthalene	91-20-3	5.85 (2.3)	29.36 (10.42)	7.87 (3.1)	8.18 (4.1)		10.61 (4.52)	11.72 (5.7)	8.84 (4.13)	2.17 (1.3)	3.74 (2.16)	10.12 (3.12)	10.96 (4.23)			17.83 (10)	23.78 (5.15)
1,2-Naphthoquinone	524-42-5					1.45 (0.8)	0.46 (0.14)										
4-Cyanostyrene	3435-51-6	3.27 (1.35)		1.24 (0.6)	1.28 (0.97)	0.22 (0.1)		1.09 (0.85)	0.84 (0.23)				0.84 (0.5)				
4-Aminostyrene	1520-21-4				0.19 (0.1)	3.36 (1.32)											
Quinoline	91-22-5	1.67 (1.24)	3.21 (1.35)	2.95 (1.35)	5.65 (2.62)	1.43 (0.55)			0.15 (0.1)								
1,3-Phenylene diisocyanate	123-61-5					1.04 (0.64)	0.46 (0.28)	2.44 (1.2)	6.77 (3.48)						2.54 (3.12)	4.67 (3.14)	1.32 (0.97)
Phthalonitrile	91-15-6	0.51 (0.25)	0.38 (0.18)	6.31 (2.45)	1.74 (0.9)		5.49 (2.78)	2.67 (1.4)	1.45 (0.78)								
Indole	120-72-9		2.91 (1.19)	2.19 (1.1)	12.43 (5.68)												
Quinaldine	91-63-4			0.91 (0.35)	0.67 (0.2)	1 (0.4)			0.99 (0.45)				0.96 (0.4)				
1-Methylnaphthalene	90-12-0	0.80 (0.3)	2.31 (1.2)	0.91 (0.4)	1.07 (0.6)	0.87 (0.35)		1.69 (0.9)	1.38 (0.45)								
Tri(propylene glycol) propyl ether	96077-04-2													0.69 (0.4)			
Benzenebutanenitrile	2046-18-6									1.71 (0.96)		2.59 (1.18)	0.33 (0.1)				
Naphthalene, 1-azido-	6921-40-0			0.42 (0.18)	0.86 (0.36)			0.54 (0.25)	0.81 (0.21)								
2,4-Diisocyanato-1-methylbenzene (TDI)	584-84-9	25.10 (10.1)	2.68 (1.22)	10.71 (5.78)	1.43 (0.75)	7.79 (2.45)	2.65 (1.25)	3.24 (1.38)	1.87 (0.95)	13.58 (7.14)	2.54 (1.25)		1.56 (0.75)	7.77 (5.1)			
Biphenyl	92-52-4		1.95 (0.85)	0.59 (0.48)	2.28 (1.17)	1.04 (0.6)	1.48 (0.78)	2.34 (1.1)	2.95 (1.3)			1.22 (0.65)	2.89 (1.34)			1.27 (0.65)	2 (0.84)
Naphthalene, 2-ethenyl-	827-54-3		0.86 (0.4)	0.21 (0.1)	1.77 (1.18)	0.9 (0.25)	1.14 (0.6)	0.74 (0.29)	0.92 (0.4)								
Acenaphthylene	208-96-8		17.02 (8.09)	0.93 (0.44)	10.26 (6.8)	0.75 (0.21)	5.06 (2.3)	5.73 (2.4)	8.02 (3.98)	0.32 (0.22)		1.32 (0.7)	5.02 (2.32)			4.69 (2.14)	12.49 (6.42)
Naphthalene, 1-isocyano-	1984-04-9		2.89 (1.19)	1.35 (0.28)	12.62 (7.11)	0.47 (0.18)	9.23 (4.21)	3.80 (1.34)	5.58 (2.21)	0.34 (0.39)		1.04 (0.62)	2.74 (1.2)			6.67 (3.45)	6.24 (3.1)
Dibenzofuran	132-64-9						1.32 (0.65)	0.39 (0.2)	3.69 (1.2)							0.79 (0.5)	0.94 (0.21)
2,5-Cyclohexadiene-1,4-dione, 2,6-bis(1,1-dimethylethyl)-	719-22-2									0.66 (0.2)							
2-Naphthalenecarbonitrile	613-46-7		1.76 (0.8)	0.61 (0.31)	11.11 (5.2)	0.66 (0.2)	6.78 (3.6)	2.32 (1.21)									
Fluorene	86-73-7		2.36 (1.51)		0.49 (0.2)	2.03 (1.2)	4.56 (2.24)	0.69 (0.32)	2.11 (1.1)				1.05 (0.76)				
Anthracene	120-12-7		2.76 (0.95)		6.40 (2.3)	0.40 (0.12)	1.78 (0.98)	0.42 (0.18)	0.35 (0.2)							1.86 (0.86)	2.82 (1.2)
Tris(1-chloro-2-propyl)phosphate (TCPP)	13674-84-5													19.71 (5.28)			

## Data Availability

Not applicable.

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
