# Peer review of "Analysis of Flammability and Smoke Emission of Plastic Materials Used in Construction and Transport"

_materials, 2023, doi:10.3390/ma16062444_

Round 1

Reviewer 1 Report

The manuscript deals with evaluation of fire toxicity of commercial PU foams. To be considered for publication, the manuscript needs to be modified according to the following suggestions:

1)     The English of the manuscript needs thorough editing, especially, there are numerous typos

2)     Tube furnace is known for almost a decade now and has been used extensively to analyze the PU foams. The authors need to summarize the already published results and  justify this study.

3)     Though, the PU foams are commercials, one needs to provide details about is regarding its chemistry and composition. As, most likely flame retardants are present in the foams, elemental composition using EDX analysis should be provided. 

4)     Can tube furnace results be comparable to results obtained using Cone-FTIR or smoke density chamber?

5)     Provide error values for all data provided toxic gas evaluation

6)     Recent efforts on complete evaluation of toxicity of  smoke produced during combustion of flame retardant polymers should be cited and included in the introduction

Eg. https://doi.org/10.1016/j.impact.2022.100414

Author Response

Thank the Reviewer 1 for all your valuable comments. We agree with your recommendations that the manuscript should be improved. We carefully improved the manuscript with the recommendations point by point:

Point 1: The English of the manuscript needs thorough editing, especially, there are numerous typos

Response 1: English language of the manuscript was improved.

Point 2: Tube furnace is known for almost a decade now and has been used extensively to analyze the PU foams. The authors need to summarize the already published results and  justify this study.

Response 2: W searched the literature and find the articles related with tube furnace and polyurethane materials. We summarized already published articles and we presented results in one of the paragraphs in the Introduction. We think our work is interesting because it is hard to find works that contain comprehensive information of fire toxicity of polyurethane materials. In most works, only inorganic gases are determined. Available literature is also focused on products which are released during decomposition and combustion taking place under only one or two selected measurement conditions, for example: during pyrolyzed at 950 °C or combustion at 600 °C. Our study are focused on toxicity of fire effluents from real fires under different fire scenarios: oxidative pyrolysis, well ventilated flaming, small vitiated fires and post-flashover fires.

Point 3: Though, the PU foams are commercials, one needs to provide details about is regarding its chemistry and composition. As, most likely flame retardants are present in the foams, elemental composition using EDX analysis should be provided.

Response 3: We agree that chemistry and composition of PUR foams would enrich the work. We have conducted an EDX analysis of the foams, but the results are unsatisfactory and they contribute nothing to our work. That is why we did not include them in the article. However, we include them in the appendix.

However, we provided the densities of the tested foams in the article in Materials

Point 4: Can tube furnace results be comparable to results obtained using Cone-FTIR or smoke density chamber?

Response 4: It is hard to compare the results obtained using Cone-FTIR or smoke density chamber-FTIR and tube furnace-FTIR. In the smoke chamber the combustion conditions are varying during the course of the combustion as the smoke gases produced fill the closed chamber, whereas the combustion conditions in a cone calorimeter test are always well ventilated using the normal test procedure. While tube furnace used to model a wide range of fire conditions. However, the remark is very interesting, and maybe in future works we will try to carry out such works.

Point 5: Provide error values for all data provided toxic gas evaluation

Response 5:  Unfortunately, we cannot provide error values for data provided toxic evaluation at the moment. However, we believe that providing these values will not significantly affect the meaning of the article.

Point 6: Recent efforts on complete evaluation of toxicity of  smoke produced during combustion of flame retardant polymers should be cited and included in the introduction

Eg. https://doi.org/10.1016/j.impact.2022.100414

Response 6: Of course now we included recent efforts on complete evaluation of toxicity of  smoke produced during combustion of flame retardant polymers in the Introduction.

Reviewer 2 Report

1.             The references need revision for consistency

2.             Page 11: Check and revise the caption of the y-axis of figure 5 (hexane and ethane emission)

3.             Page 3: Formaldehyde (CHOH) and HCOH in figure 4 should be revised to HCHO.

4.             Page 3: Materials: Give more details about foam density

5.             Did the authors control mass of foam before Burning test?

6.             Table2: Two columns contain V5 and T5. Improve them.

7.             Page 4, Depending on the tested foams, the occurrence of two to four stages of decomposition corresponds to the maximum rate of degradation of hard segments (T1, T2, T3) and soft segments (T4, T5, T6). Why does the PUF_D have only the maximum rate of degradation of hard segments (T1 and T2)? Where are the maximum rate of degradation of soft segments?

8.             Page 10 Figure 5: the authors mentioned that the yields of C2H4 increases with the temperature from 650 to 825 °C. Why did PUF_D show the lowest yield of C2H4?

9.             Page 4: the loss of weight (TG) and derivative of the loss of weight (DTG) should be revised. Loss of weight is not TG.

Author Response

Dear Reviewer 2,

We have modified the manuscript accordingly and the detailed corrections with your recommendations. We improved quite the large number of other minor mistakes that we noticed while working on the text. We believe that our manuscript in the present form can be published in Materials.

Changes in the text are marked in red.

Sincerely,

Authors

Round 2

Reviewer 1 Report

The English of the manuscript is still inappropriate.

It needs to be edited by a native speaker. I don't agree with the authors regarding the error values. In any analytical measurement, error values are needed to determine the accuracy of the results. One has to also mention how many measurements were done for each sample. Thus I suggest they be included in the revised manuscript.

Author Response

Thank the Reviewer 1 for all your valuable comments. We agree with your recommendations that the manuscript should be improved. We carefully improved the manuscript with the recommendations:

Point 1: It needs to be edited by a native speaker.

Response 1: English language of the manuscript was improved by MDPI native speaker.

Point 2: I don't agree with the authors regarding the error values. In any analytical measurement, error values are needed to determine the accuracy of the results. One has to also mention how many measurements were done for each sample. Thus I suggest they be included in the revised manuscript.

Response 2: We added the error values of our results and included the information how many measurements were done for each sample. The values presented in article are the averages obtained for at least three samples from each sample.

We hope that everything is now clear and the article can get a positive review.

Reviewer 2 Report

The authors have revised the articles satisfactorily.

However, the following revisions are required

1. Page 2: W. Netkueakul et all. [17] have investigate

The authors need to revise this as "W. Netkueakul et al. [17] have investigated"

2. A similar revision is required for Singh et all. [18] on the same page.

Author Response

Thank the Reviewer 2 for all your valuable comments. We agree with your recommendations that the manuscript should be improved. We carefully improved the manuscript with the recommendations:

Point 1: 1. Page 2: W. Netkueakul et all. [17] have investigate

The authors need to revise this as "W. Netkueakul et al. [17] have investigated"

  1. A similar revision is required for Singh et all. [18] on the same page.

Response 1: Of course we improved that sentences. Moreover, the English language of the manuscript was improved by MDPI native speaker.